# The Microtubule Cytoskeleton during the Early *Drosophila* Spermiogenesis

**DOI:** 10.3390/cells9122684

**Published:** 2020-12-14

**Authors:** Maria Giovanna Riparbelli, Veronica Persico, Giuliano Callaini

**Affiliations:** 1Department of Life Sciences, University of Siena, Via Aldo Moro 2, 53100 Siena, Italy; riparbelli@unisi.it (M.G.R.); persico@student.unisi.it (V.P.); 2Department of Medical Biotechnologies, University of Siena, Via Aldo Moro 2, 53100 Siena, Italy

**Keywords:** *Drosophila*, spermiogenesis, microtubules, dense complex, centrosomal proteins

## Abstract

Sperm elongation and nuclear shaping in *Drosophila* largely depends on the microtubule cytoskeleton that in early spermatids has centrosomal and non-centrosomal origins. We report here an additional γ-tubulin focus localized on the anterior pole of the nucleus in correspondence of the apical end of the perinuclear microtubules that run within the dense complex. The perinuclear microtubules are nucleated by the pericentriolar material, or centriole adjunct, that surrounds the basal body and are retained to play a major role in nuclear shaping. However, we found that both the perinuclear microtubules and the dense complex are present in spermatids lacking centrioles. Therefore, the basal body or the centriole adjunct seem to be dispensable for the organization and assembly of these structures. These observations shed light on a novel localization of γ-tubulin and open a new scenario on the distribution of the microtubules and the organization of the dense complex during early *Drosophila* spermiogenesis.

## 1. Introduction

Microtubules (MTs) play a central role during *Drosophila* spermiogenesis and are mainly involved in the elongation of the spermatids and the assembly of the motile sperm axoneme. MTs of the sperm axoneme growth from the cilium-like regions are inherited at the end of spermatogenesis [1] and their elongation does not require the intraflagellar transport (IFT) machinery, which uses microtubule motors to transport cargo from the cell body to the ciliary tip and back [2]. Rather, the sperm axoneme of the *Drosophila* spermatids requires a cytoplasmic mode of assembly [3,4]. Accordingly, it has been demonstrated that the maturation of the axoneme relies on the local translation of specific key components from the cytoplasm that incorporate into the forming axoneme as it emerges from the ciliary cap [5]. The assembly and further elongation of the axonemal MTs would require tubulin translocation by diffusion through the ciliary cap. However, the observation of ribosomes within the apical region of the ciliary cap [6] suggests that protein synthesis in this region could assist in sustaining MT elongation.

The centriole that organizes the sperm axoneme is associated in young spermatids with a low amount of pericentriolar material from which a small astral array of MT nucleates. After the docking of the centriole to the nuclear envelope, the amount of pericentriolar material gradually increases to form a characteristic non-membranous structure, the so-called centriole adjunct. The term centriolar adjunct in insect spermatids denotes a different structure than in mammalian spermatids, where this term indicates a structure growing from the proximal centriole and similar in morphology to the primary cilium [7,8]. The centriole adjunct in *Drosophila* is the nucleation site of two opposite bundles of MTs. One bundle, the perinuclear MTs, grows towards the apical end of the nucleus; the other bundle, the tail MTs, extends within the spermatid length [9]. However, neither the axoneme MTs, nor the tail MTs, seem to be involved in the elongation of the spermatids. Flies’ mutants for genes involved in the assembly of the centrioles, such as Sas-4, lack axonemes and centriole adjunct, but their spermatids are able to elongate, even if they never reach the full length observed in wild-type sperm [10].

It has been suggested that spermatid elongation in *Drosophila* requires the cooperation between cytoplasmic non-axonemal MTs and the mitochondrial derivatives that might provide a structural scaffold for MT reorganization, as inferred by the finding of γ-tubulin spots on the mitochondrial membrane [11]. Accordingly, it has been shown that a splice variant of centrosomin associates with the mitochondrial membrane to form distinct speckles that recruit γ-tubulin and regulate MT assembly [12].

The MTs that nucleate from the centriole adjunct have been involved in the shaping of the spermatid nucleus [6,13]. They acquire an asymmetric disposition at the beginning of the spermatid differentiation, and most of them form a longitudinal bundle along a side of the elongating nucleus. Such MTs are enclosed within an electron-dense material, the so-called dense complex [9,14], that also contains filamentous actin [15] and the actin-binding protein anillin [13]. The dense complex is firstly found in young spermatids as a hemispherical thin cup of electron-dense material on the side of the nucleus opposite the nebenkern. Concurrently to the nuclear elongation, the dense body increases dimension and shifts to the lateral side of the nucleus where the nuclear pores are concentrated (Figure 1, [13,14]).

Proteins, such as Spag4 and yuri gagarin (yuri), that localize respectively on the inner nuclear envelope and along the dense body [15,16,17], could be involved in linking the MT network, the main actor of nuclear shaping, to the nuclear membrane. A third protein, Salto, is distributed on the dense complex [18] and its mutations phenocopy *Spag* and *yuri* mutants, suggesting that all these proteins are required for sperm head morphogenesis. All these observations suggest that the MT framework plays a pivotal role during spermatid morphogenesis and differentiation in *Drosophila*.

We show, here, that an additional MT organizing center is present in the apical region of the spermatid nuclei in correspondence of the distal end of the MTs that run within the dense complex. Moreover, we also report that a dense complex is also present in elongating spermatids lacking basal bodies, suggesting that the basal body is dispensable for the organization of this structure.

## 2. Materials and Methods

### 2.1. Drosophila Strains

OregonR stock was used as control. The stock containing the Uncoordinated-GFP (Unc-GFP) [19] and Pericentrin-like protein-GFP (Pact-GFP [20]) transgenes were provided by M.J. Kernan and J. Raff, respectively. The *Sas-4* [10,21] and *Cog7* [22] mutant flies were kindly provided by J. Raff and M.G. Giansanti, respectively. Flies were raised on a standard *Drosophila* medium in 200 mL plastic containers at 24 °C. Testes were dissected from older pupae, pharate adults, and just emerged males.

### 2.2. Antibodies and Reagents

Nocodazole, Latrunculin A, Dimethyl sulfoxide (DMSO), and Shields and Sang M3 Insect Medium were purchased from Sigma-Aldrich (St. Louis, MO, USA). Nocodazole was dissolved at 1 mg/mL in DMSO and stored frozen at −20 °C. We used the following antibodies: rat anti-α-tubulin-YL1/2 (1:50; ab6160 Abcam), mouse anti-acetylated tubulin (1:100; T7451 Sigma-Aldrich), mouse anti-α-tubulin (1:500; T5168 Sigma-Aldrich), mouse anti-γ-tubulin GTU88 (1:100; T6557 Sigma-Aldrich), mouse anti-polyglutamylated tubulin GT335 (1:200; T9822 Sigma-Aldrich), rabbit anti-centrosomin (Cnn, 1:400; [23]), rabbit anti-Spd-2 (1:500; [24]), and rabbit anti-pavarotti (Pav-KLP, 1:400; [25]). The secondary antibodies used (1:800), Alexa Fluor 488, 555, and 647 anti-mouse, anti-rabbit, anti-rat, and anti-chicken, were obtained from Invitrogen. Hoechst 33,258 was obtained by Sigma-Aldrich.

### 2.3. Culture and Drug Treatment Experiments

Testes were dissected from pupae between 5 and 7 days after pupation. At this stage, testis contain cysts at the spermatogonial, spermatocyte, and spermatid stages. Testes were then transferred in a 200 μL of M3 medium into a sterile 24-well plate. To assess the effect of MT depolymerization, the dissected testes were exposed to cold for 10 min or cultured at 24 °C for 24 h in M3 medium containing 100 nM Nocodazole. To inhibit microfilament polymerization, testes were incubated at 24 °C for 24 h in M3 medium containing 100 nM Latrunculin A. Specimens were fixed and stained following 24 h drug or vehicle treatments. The experiments were repeated three times.

### 2.4. Immunofluorescence Preparations

Testes were squashed under a small cover glass and frozen in liquid nitrogen. After removal of the coverslip, the samples were immersed in methanol for 10 min at −20 °C. For localization of MTs and centriole/centrosome components, the fixed samples were washed for 15 min in PBS and incubated for 1 h in PBS containing 0.1% bovine serum albumin (PBS–BSA). The samples were incubated overnight at 4 °C with the specific antisera and then with anti-tubulin antibodies for 4–5 h at room temperature. After washing in PBS–BSA, the samples were incubated for 1 h at room temperature with the appropriate secondary antibodies. For simultaneous localization of γ-tubulin and α-tubulin, testes were incubated in either GTU88 or YL1/2 antibodies followed by anti-mouse or anti-rat secondary antibodies. DNA was visualized with incubation of 3–4 min in Hoechst. Testes were mounted in small drops of 90% glycerol in PBS.

### 2.5. Image Acquisition

Images were taken using an Axio Imager Z1 (Carl Zeiss, Jena, Germany) microscope equipped with an HBO 50-W mercury lamp for epifluorescence and with an AxioCam HR cooled charge-coupled camera (Carl Zeiss). Gray-scale digital images were collected separately with the AxioVision 2.1 software (Carl Zeiss) and then pseudocolored and merged using Adobe Photoshop 7.0 software (Adobe Systems).

### 2.6. Transmission Electron Microscopy (TEM)

Testes were fixed overnight at 4 °C in 2.5% glutaraldehyde in PBS. After rinsing for 30 min in PBS, the material was post-fixed in 1% osmium tetroxide in PBS for 2 h. The samples were dehydrated in a graded series of ethanol, and then infiltrated with a mixture of Epon–Araldite resin and polymerized at 60 °C for 48 h. Ultrathin sections were cut with a Reichert ultramicrotome, collected with formvar-coated copper grids, and stained with uranyl acetate and lead citrate. TEM preparations were observed with a Tecnai G2 Spirit EM (FEI) equipped with a Morada CCD camera (Olympus).

## 3. Results

### 3.1. MT Distribution Significantly Changes during Early Drosophila Spermiogenesis

Since MTs represent an important actor during early morphogenesis of the *Drosophila* spermatids, we examined the distribution of these cytoskeletal organelles at different stages of early spermiogenesis (Figure 1) to better understand their role in nuclear shaping and tail elongation. The round *Drosophila* spermatids at the onion stage display a large cytoplasmic MT network that surrounds the nuclei and mitochondria and a small astral array of MTs close to the basal region of the nucleus (Figure 2A). At the beginning of axoneme elongation, distinct bundles of MTs, the tail MTs, are emanated from the aster close to the nucleus and run parallel through the cytoplasm of the spermatid (Figure 2B,B’). Other MTs, the perinuclear MTs, originated from this aster and surrounded the nucleus (inset, Figure 2B’). The parallel bundles of tail MTs compacted (Figure 2C) concurrently to the reduction of the spermatid diameter during further elongation and most of the perinuclear MTs formed a large bundle at one side of the nucleus at the leaf stage (Figure 2C,C’). Perinuclear and tail MTs were separated at the proximal end of the nucleus by a distinct gap where the basal body was located (Figure 2C’).

The anti α-tubulin antibody recognized all the cytoplasmic MTs, thus hindering the axoneme that represents an important actor during the process of spermatid differentiation. Therefore, we stained *Drosophila* testes expressing Uncoordinated (Unc)-GFP, a potential ortholog of the orofaciodigital syndrome 1 (Ofd1) that is required for ciliogenesis and mechanosensation in *Drosophila* and localizes to the basal body [19], with an antibody against acetylated tubulin to recognize both centrioles and stable MTs associated with the axonemes. Unlike mammalian cells, in which stable MTs are only associated with ciliary axonemes, the anti-acetylated tubulin antibody recognized some cytoplasmic MTs in *Drosophila* testis in addition to the axoneme MTs (Figure 3A,B).

One feature that is unique to meiosis II in *Drosophila* and insect spermatogenesis is the presence of a single centriole in each centrosome [26,27]. Therefore, at the end of meiosis, each spermatid inherits a centriole and an associated cilium-like region (CLR) (Figure 3A″) that is the precursor of the sperm axoneme [1]. The small asters of MTs (Figure 3A’), also seen with the antibody against α-tubulin (Figure 2A), were found in correspondence of the proximal region of the basal bodies (Figure 3A″). The Unc protein is involved in centriole-to-basal body conversion and is specifically expressed by spermatocyte and spermatid centrioles that nucleate a ciliary axoneme [19]. We found that this protein was localized in three distinct regions of the basal bodies (Figure 3A″): a larger proximal region, corresponding to the basal body, an intermediate bright spot region, matching the transition zone, and a distal region corresponding to the CLR. The small aster of MTs was still evident at the beginning of spermatid elongation (Figure 3B,B’,C,C’) when both the intermediate and the distal Unc-GFP signals moved away concurrently to the axoneme growth (Figure 3B,B″,C,C″). The intermediate Unc-GFP spot was identified as the ring centriole at the basis of the ciliary cap in which the axoneme assembled [13]. In more elongated spermatids, the small aster was no longer visible, but distinct MTs formed a shell around the nucleus (Figure 3D,D’). The perinuclear MTs reorganized at the leaf and canoe stages in distinct parallel bundles that run at one side of the nuclear envelope from the basal body to the apical region of the nuclei (Figure 3E,E’). The uniform Unc-GFP signal at the basal body changed significantly during the process of spermatid elongation. Specifically, we observed a proximal Unc-GFP signal that was weak at the beginning of axoneme elongation (Figure 3B″) and became larger and abnormal in shape as differentiation proceeded (Figure 3C″,D″). A round Unc-GFP signal was also seen at the proximal end of the axoneme, whereas a pair of bright signals were found in the medial-distal region of the basal body (Figure 3C″,D″). The Unc-GFP staining localized at the canoe stage in an irregular dense cluster associated to both perinuclear and tail MTs (Figure 3E″).

To assess MT stability during early spermiogenesis, we treated *Drosophila* testes with the tubulin-sequestering drug nocodazole and with cold temperature. Nocodazole has the advantage of being more specific for MTs, whereas cold temperature is likely to affect other cellular processes [28]. Both treatments produced the loss of cytoplasmic and ciliary cap MTs, suggesting that the majority of the cytoplasmic MTs recognized by the antibody against acetylated tubulin were less stable. Remarkably, the distal end of the axonemal MTs detached from the ring centrioles, and frayed after nocodazole incubation (Figure 4A), whereas cold temperature did not affect the axoneme integrity (Figure 4B). The different results could be explained by factors that stabilize MTs against cold-induced depolymerization at the ring centrioles, but more sensitivity to drug treatment, thus resulting in frayed distal ends of the axonemes.

Likewise, polyglutamylated tubulin is absent from cytoplasmic MTs, but it is restricted to the spermatid axonemes (Figure 4C,D). Perinuclear and tail MTs did not contain polyglutamylated tubulin.

It has been suggested that the dense complex where the perinuclear MTs formed a continuous bundle also contains actin [15]. Recent studies have shown that the nucleation of MTs and actin filaments is closely related, and that the centrosome can also be the nucleation center of actin filaments [29,30]. Thus, we decided to verify the role of the microfilaments in the assembly/maintenance/dynamics of the dense complex by inhibiting actin polymerization and disrupting microfilament organization with Latrunculin A. We found that Latrunculin incubation did not substantially impair nuclear elongation but led to a significant increase in length of the MTs associated with the dense body that extended the length of the nuclei. We found that the perinuclear MT bundles measured 12.97 ± 01.4 μm in control canoe stage spermatids (Figure 4E, *n* = 47) and reached 23.61 ± 0.21 μm in drug-treated canoe stage spermatids (Figure 4F, *n* = 38).

### 3.2. γ-Tubulin is Present in Distinct Foci during Early Spermiogenesis

Due to the main role of γ-tubulin in MT nucleation [31,32], we also look at the localization of this protein during spermatid differentiation. γ-tubulin overlapped the whole basal body in young round spermatids (Figure 5A–A″) and this localization persisted at the beginning of axoneme elongation (Figure 5B–B″). Two small spots of γ-tubulin, in addition to the labelling at the basal body, were also seen when the axoneme underwent elongation: one spot corresponded to the ring centriole, whereas the other was located on the anterior pole of the nucleus in correspondence of the acrosomal region (Figure 5B–B″). The γ-tubulin spot localized on the anterior part of the nucleus persisted in elongating spermatids during the leaf (Figure 5C–C″) and canoe (Figure 5D–D″) stages, whereas the γ-tubulin associated to the proximal region of the nucleus transformed in a distinct ring around the basal body (Figure 5C,C’,D,D’). Double labelling of leaf (Figure 5E–E″) and canoe (Figure 5F–F″) stage nuclei with antibodies against α- and γ-tubulin showed that the larger γ-tubulin ring was associated with the basal region of the axoneme, whereas the smaller spot was localized at the apical end of the perinuclear MTs.

We then wondered whether the γ-tubulin accumulation persisted in young spermatids after depolymerization of the cytoplasmic MTs. Thus, we counterstained nocodazole-treated spermatids expressing Unc-GFP with an antibody against γ-tubulin. We found that γ-tubulin accumulation persisted at the basal bodies and as small dots at the apical side of the spermatid nuclei (Figure 6A). Therefore, the localization of γ-tubulin during early spermiogenesis did not strictly depend on MT integrity.

The expansion of the pericentriolar material and the recruitment of γ-tubulin at the centrosome depends on fly embryos by a feedback loop in which the conserved proteins Spindle defective 2 (Spd-2) and Centrosomin (Cnn) play crucial roles [33,34,35,36,37]. We wondered, therefore, whether the localization of γ-tubulin during early spermiogenesis may depend on these proteins. Spd-2 was localized along the whole centriole during spermatogenesis (not shown) but was restricted to the proximal end of the basal body in onion stage spermatids (Figure 6B). The localization of Spd-2 to the basal body was no longer observed in early elongating spermatids, but a single spot of this protein was seen on the apical region of the nuclei (Figure 6C). Spermatid elongation is accompanied by a dramatic change in nuclear shaping, but the Spd-2 spot persisted in the apical region of the nuclei in association with the distal end of the perinuclear MTs that run within the dense complex (Figure 6D). Cnn was found at the basal bodies of onion stage (Figure 6E) and elongating (Figure 6F) spermatids, where this protein persisted in close association with the proximal end of the perinuclear MTs (Figure 6F) but was not detected at the distal ends of the perinuclear MT bundles (Figure 6F).

It has been demonstrated that a scaffold of the conserved Pericentrin-like protein (Plp) and Cnn is required for centrosome function in fly embryos [38,39]. Moreover, Plp helps assemble the electrodense pericentriolar clouds that appear to organize the MTs around the centriole during *Drosophila* male gametogenesis [40]. Thus, we ask whether the localization of Plp could reveal additional MT nucleating sites during spermatid differentiation. We then stained *Drosophila* testes expressing the Pact-GFP portion of Plp [20] with an antibody against tubulin. Pact-GFP localized to basal bodies in young onion stage spermatids (not shown) and this localization persisted during axoneme elongation (Figure 6G). However, Pact-GFP was not found in the apical region of the nuclei.

### 3.3. The Centriole Adjunct is the Main Regulator of Perinuclear MTs

Young wild-type spermatids at the onion stage displayed a characteristic thin cluster of dense material, the dense complex, enclosed between the nuclear envelope and a flat cisterna of endoplasmic reticulum [13,14]. The dense complex surrounds the basal region of the nucleus in front of the basal body (Figure 7A) and contains thin filaments (5–7 nm thick) that run parallel to the nuclear envelope or link the nuclear pores to the surrounding cisterna of endoplasmic reticulum (inset, Figure 7A). At the onset of spermatid differentiation, the basal bodies had a random orientation and were not in contact with the nuclear envelope (Figure 7B). Many MTs nucleated from the proximal region of the basal bodies and some of them interacted with the dense complex (Figure 7B). Such MTs likely assist basal body docking to the nuclear envelope. After the docking of the basal body to the nuclear envelope, most of the MTs that nucleated from the proximal region of the basal body were integrated within the dense complex to form a perinuclear sheath (Figure 7C). Remarkably, some MTs were present within the dense complex before the basal body contacted the nuclear envelope. The dense material that surrounded the basal body increased in elongating spermatids at the leaf stage and formed a compact cluster, called the centriole adjunct (Figure 7D). The centriole adjunct reached its maximum extension at the early canoe stage and then began to contract following the displacement of the basal body within the nuclear cavity ([6]; Figure 7E). The proximal region of the centriole adjunct was the anchoring site for the two mitochondrial derivatives resulting from the unwrapping of the nebenkern (Figure 7E). Two distinct MT bundles nucleated in opposite directions from the centriole adjunct (Figure 7D,E): one bundle runs through the cytoplasm of the elongating spermatid until its distal end, whereas the other MTs clustered within the dense complex at one side of the nucleus. Cross-sections of the dense complex showed tightly packed MTs scattered within an electron-dense fibrous material (Figure 7F). We examined 34 cross-sections of dense complexes from different canoe stage spermatids, scoring a total of 2386 single MTs. We found adjacent MTs linked together by thin bridges and single MTs bearing short projections ending within the fibrous material (Figure 7F). We did not observe MTs linked to the nuclear envelope.

### 3.4. The Centriole Adjunct is Dispensable for the Formation of the Dense Complex

There are several examples of *Drosophila* mutants in which the meiotic divisions failed, thus leading to abnormal inheritance of nuclei and mitochondria derivatives in early spermatids. Therefore, it is possible to find two nuclei in the same spermatid, one axoneme and one nebenkern (Figure 7G). Remarkably, despite the presence of only one basal body, both the nuclei displayed normal looking dense complexes, suggesting that the assembly of these structures does not require functional basal bodies or centriole adjuncts. Moreover, during normal spermiogenesis, the dense complex forms close to the basal region of the nucleus before the basal body docks to the nuclear envelope (Figure 7B). We, therefore, asked whether the basal bodies and their associated centriole adjuncts may have some impact on the formation and the spatial localization of the dense complex.

*Sas-4* mutant spermatocytes have rare centrioles [21], thus representing a good system in which to analyze the relationship between MT organization and spermatid differentiation. Although the spermatogonial divisions occur in the absence of centrioles, the larger meiotic cells are unable to assemble functional bipolar spindles unless a complete centriole pair is present (Figure 8A) and the meiotic divisions fail. However, the abnormal spermatids undergo some differentiation processes such as chromatin condensation, formation of irregular mitochondrial derivatives, and partial elongation (Figure 8B). The ring canals detected by an antibody against the mitotic kinesin 6 protein encoded by Pavarotti (Pav-KLP) were clustered at the distal end of the elongating spermatids as it occurs during wild-type spermiogenesis (Figure 8C). Distinct MTs run the whole length of the elongating spermatids, but the nuclei were scattered along the spermatid length (Figure 8B) rather than being aligned at the apical end of the spermatid cyst as it occurs in control testes (not shown). Careful analysis revealed distinct MT bundles localized at one side of the nuclei in elongating spermatids (Figure 8D). Such MTs were short and ended just beyond the nuclei (inset, Figure 8D). Since these MT bundles look like those observed within the dense complexes in wild-type leaf and canoe stage spermatids, we asked whether a dense complex could also be present in *Sas-4* mutant testes.

Ultrastructural analysis of *Sas-4* onion stage mutant spermatids showed that a distinct dense complex in the form of a hemispherical cup surrounded the round nuclei in front of the nebenkern (Figure 9A). As the spermiogenesis progressed, some nuclei underwent elongation, although they appeared variable in size and shape (Figure 9B,C). The nuclear pores clustered at one side of the abnormal nuclei and the dense complex was restricted to the fenestrated area where the chromatin condensed (Figure 9B,C). Despite the absence of basal bodies, we observed that several MTs ran through the dense bodies parallel to the fenestrated side of the nuclear envelope (Figure 9B’,C’). Serial longitudinal sections of leaf stage nuclei (Figure 9D–D″’,E–E″’) showed that 22 of the dense complexes examined in different spermatids (*n* = 53) contained one or occasionally two hollow irregular spheres of electron-dense material (Figure 9D’,D″,E’,E″). Similar hollow spheres had also been observed near the distal region of 15 of the leaf stage nuclei examined (*n* = 53) (Figure 9F). Many MTs radiated from the wall of the hollow spheres and ran parallel to the nuclear envelope (Figure 9F).

We then wondered if the perinuclear MTs observed in *Sas-4* mutant spermatids were nucleated in the absence of basal bodies by γ-tubulin foci. We found, near the nuclear region, distinct spots of γ-tubulin (Figure 9G,H) that appear at higher magnification as ring-shaped structures (insets, Figure 9G,H). This observation suggests that the γ-tubulin spots may correspond to the spherical aggregates of dense material observed by electron microscopy.

### 3.5. The Nucleoporin Nup154 is Restricted to the Dense Complex

The Nup154, a *Drosophila* nucleoporin related to the vertebrate Nup155, essential for male and female gametogenesis [41], was uniformly localized as a finely punctate staining around the whole nuclear envelope of primary and secondary spermatocytes (Figure 10A). In young primary spermatocytes, at the onion stage, the Nup154 staining was restricted to the proximal side of the nuclear envelope opposite the basal body (Figure 10B). Nup154 formed a cluster at one lateral side of canoe stage nuclei (Figure 10C) but was no longer detected in mature sperm (not shown).

## 4. Discussion

The differentiating *Drosophila* spermatid represents a unique model in which centrosomal and non-centrosomal MT organizing centers coexist [42]. The spermatids have a basal body that nucleates the axonemal MTs and recruits the material that forms the centriole adjunct, the master regulator of tail and perinuclear MTs. In addition, distinct MTs originate close to the outer membrane of the mitochondrial derivatives [11] where speckles of a splice variant of Cnn (CnnT) able to recruit γ-TuRCs have been detected [12].

Here, we reported that an additional γ-tubulin cluster is present on the anterior pole of the nucleus opposite to the basal body. This cluster is visible from the onion to the leaf/canoe stages but is no longer detected in mature sperm. A similar γ-tubulin dot has also been observed in elongating spermatids of the coleopteran *Tenebrio molitor* [43], suggesting that this localization could represent a conserved evolutive tract of the insect spermiogenesis. The significance of this γ-tubulin localization is unclear. One possibility could be that γ-tubulin may help to organize a portion of the perinuclear MTs. This protein has been observed before the docking of the basal body to the nuclear membrane, supporting the possibility that a distinct class of the perinuclear MTs might be organized independently by the centriole adjunct. If it is so, the perinuclear MTs could have two different origins, from the centriole adjunct and from the apical γ-tubulin cluster, thus running in opposite directions. This hypothesis explains the presence of MTs within the dense complex before the docking of the basal body to the nuclear membrane, and the finding of a few MTs that go beyond the basal body region and run towards the posterior region of the elongating spermatids. The involvement of the small γ-tubulin cluster in MT nucleation is also supported by the finding that the centrosome-associated protein Rb188 is associated with the apical end of the leaf stage nuclei [44]. Spd-2, a protein involved in centrosome maturation [35], is also found at the apical end of the spermatid nuclei in the region where γ-tubulin is localized. In contrast, Cnn, that works with Spd-2 to ensure centrosome maturation during mitosis [34,37], is only associated to the centriole adjunct. This suggests that Spd-2 alone is sufficient to ensure the proper recruitment of the apical γ-tubulin cluster. This finding is in apparent contrast with the observation that CnnT speckles help to recruit γ-tubulin at the mitochondrial membrane during spermatid elongation [12]. However, the CnnT isoform has not been detected within the centriole adjunct and at the apical end of the nuclei [12]. Salto and yuri, two proteins involved in the assembly and organization of the dense complex, have also been found in the apical region of the elongating spermatids [15,18]. Therefore, the localization of γ-tubulin in this region might support a role of this protein in the nucleation of some of the MTs that fill the dense complex.

Alternatively, the small γ-tubulin cluster close to the anterior region of the spermatid nuclei could be involved in the nucleation of a transient MT array required for the biogenesis of the acrosome. The acrosome seems to play a role during nuclear shaping in mice spermatids [45,46]. Mutations in *Merlin*, the homolog of the human *Neurofibromatosis 2 (NF2)* gene, affect *Drosophila* spermatogenesis at multiple stages and elongating spermatids display strong nuclear shaping defects [47]. Remarkably, the *merlin* gene product was also found as a bright punctate dot in the acrosomal region.

It has been suggested that the elongation of the spermatids in *Drosophila* does not depend on the axoneme, as inferred by the observation that spermatids with abnormal mitochondria morphogenesis undergo partial elongation, but it relies on the coordinate action of cytoplasmic MTs and mitochondria dynamics [11]. The lateral bridges that link cytoplasmic MTs together and with the external mitochondrial membrane support a dynamic interaction between these organelles. However, it is unclear how the short MTs nucleated in astral fashion by the γ-tubulin foci associated to the mitochondrial membrane could organize in longitudinal continuous bundles. It has been proposed that cross-linking and sliding of these (mitochondria-associated) MTs may result in a stable network that ensures elongation of the sperm tail [11]. However, it has been reported that the reduced MT assembly on the nebenkern surface in the *CnnT* mutants does not affect sperm morphogenesis that retain normal fertility and tail length comparable to wild type [12]. This suggests that additional nucleating centers may compensate the loss of MTs on the mitochondrial membrane. We found that a consistent number (*n* = 37) of the *Sas-4* mutant spermatids examined at leaf stage (*n* = 53) show dense spherical structures from which many parallel MTs nucleate. It is possible that these structures may represent alternative sites of MT nucleation in the absence of basal bodies and of centriole adjuncts. Interestingly, longitudinal tail MTs are often connected by lateral bridges and some of them are embedded in small clusters of dense material [48] that likely gives stability to these bundles to support spermatid elongation.

About 32% of the *Sas-4* spermatids scored at the ultrastructural level (*n* = 89) display nuclei that undergo partial shaping resembling abnormal leaf stages. A dense complex, although abnormal in size, is usually present at one side of the irregular nuclei and contains parallel MTs, suggesting that the basal body and the centriole adjunct are redundant for the assembly of this structure and for its localization. This is also supported by the finding that the dense complex assembles during normal spermiogenesis before the basal body contacted the nuclear envelope. Moreover, several *Drosophila* meiotic mutants display spermatids that contain two or more nuclei with dense complexes but only one basal body. Therefore, these observations could challenge the prevailing view that the perinuclear MTs nucleated by the centriole adjunct are essential for nuclear shaping. The dense complex is restricted to extension of the fenestrated side of the nuclear envelope, suggesting a close relationship between chromatin condensation, nuclear pore localization, and dense complex assembly. It has been shown that Mst77F, a protein related to linker histone 1-like protein from mammals, localizes to the nuclear dense complex during spermatid elongation, suggesting a dual role in chromatin condensation and nuclear shaping through its association with perinuclear MTs [49].

The dense complex is believed to be equivalent to the nuclear manchette of the mammalian sperm [13] in which it is involved in nuclear transport and morphogenesis [50,51]. It has been proposed that Salto, a protein localized within the dense complex, could play a role in the active transport that takes place along the MT network during nuclear shaping [18]. Here, we reported an additional evidence of the involvement of the dense complex in nuclear transport. We found, indeed, that Nup154, a *Drosophila* nucleoporin related to the vertebrate Nup155, essential for male and female gametogenesis [41], is restricted to the basal region of the onion stage spermatids and transforms in a lateral strip along the long axis of the leaf/canoe stage nuclei. Remarkably, Nup154 plays a role in the nuclear localization of components of the dpp signaling pathways [52], supporting the possibility that the dense complex could be actively involved in nuclear transport.

The robust array of perinuclear MTs that emanate from the centriole adjunct in wild-type spermatids are expected to give an important support to the elongating nuclei. However, the physical mechanism that links the perinuclear membrane with the nuclear envelope is unclear. MTs belonging to the single row lining the convex side of the nuclei are linked by thin bridges to the nuclear envelope and to the surrounding endoplasmic reticulum cisterna. By contrast, the MTs that fill the dense complex lack direct links with the nuclear envelope, raising some questions about the forces eventually exerted by these MTs to drive nuclear shaping. Rather, the longitudinal MTs seem to interact with the matrix of the dense complex. It has been suggested that the N-terminal domain of KASH proteins faces the cytoplasmic side of the nuclear envelope and interacts with microfilaments and MTs [14]. When the basal body approaches to the basal region of the nucleus, the dense complex loses its continuity and the cisterna of endoplasmic reticulum opens in front of the basal body. The microfilament meshwork that fills the dense complex must break here. The signals leading to the local interruption of the dense complex as the centriole approaches are still unclear. It has been recently shown that the docking of the basal body to the nucleus in *Drosophila* requires a mechanism that relies on the restriction of Pericentrin-like protein (Plp) and the pericentriolar material (PCM) to the proximal end of the centriole [53]. MTs that nucleate from the pericentriolar material interact with the endoplasmic reticulum and might trigger the local modification of the dense complex.

Actin [15] and the actin-binding protein anillin [13] have been reported as components of the dense complex, even if their precise localization within the dense complex has not been elucidated. We found the ultrastructural evidence of distinct microfilaments within the nuclear-dense bodies of the elongating spermatids. These microfilaments are linked to the nuclear pores and form a cage around the nuclei, suggesting that the nuclear pores could play a main role in the organization of the actin network within the dense complex. Actin forms a thin filamentous network around the nuclei until the onset of spermiogenesis, and the concentration of microfilaments at one side of the nucleus follows the nuclear pore clustering. Microfilaments have been implicated in nuclear shaping during *Drosophila* spermiogenesis [13]. Therefore, treatment with the actin inhibitor latrunculin would affect the nuclear shape of young spermatids. However, we found that this treatment does not give appreciable effects on the nuclear shaping but impacts on the MTs of the dense complex that significantly elongate. This observation points to a close relationship between the actin cytoskeleton and the regulation of the dynamics of the MTs that fill the dense complex.

We have sought to better characterize the MT cytoskeleton during different stages of the early *Drosophila* spermiogenesis. Our observations uncovered a new γ-tubulin localization at the apical end of the nucleus. This γ-tubulin focus is likely involved in the nucleation of a population of perinuclear MTs that fill the dense complex. Such γ-tubulin foci are also found in *Sas-4* mutant spermatids where they nucleate distinct bundles of perinuclear MTs. Remarkably, the *Sas-4* mutant spermatids lack basal bodies, but have distinct dense complexes, suggesting that centrioles and/or centriole adjuncts are dispensable for the assembly of this structure. Future studies aimed at addressing the regulation of MT nucleation and dynamics during early *Drosophila* spermiogenesis are required to understand whether MTs specifically contribute to nuclear shaping and sperm tail elongation.

## Figures and Tables

**Figure 1 cells-09-02684-f001:**
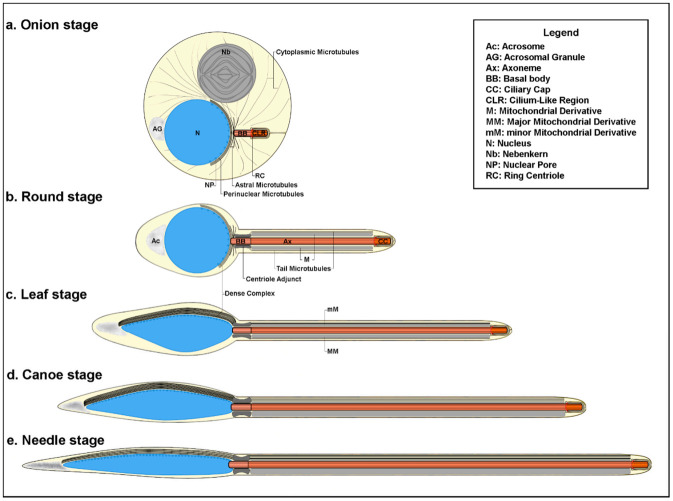
Schematic representation of the early stages of *Drosophila* spermiogenesis.

**Figure 2 cells-09-02684-f002:**
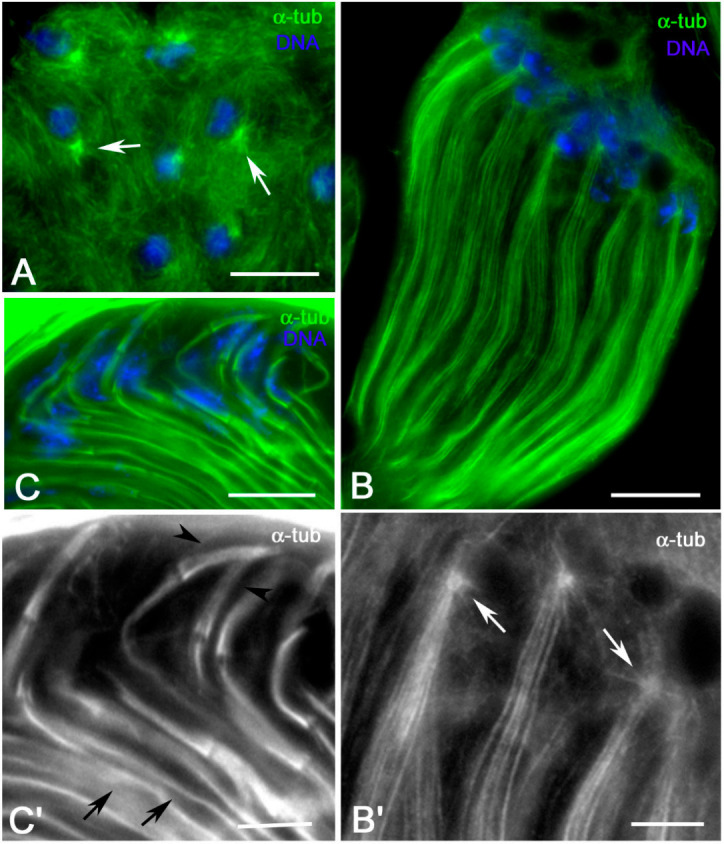
Microtubule (MT) distribution in early elongating spermatids obtained from dissected testes. (**A**) Onion stage spermatids: note the small MT asters close to the nuclei (arrows). (**B**,**B’**) Early elongating spermatids showing bundles of parallel MTs that originate from the small asters close to the nuclei (arrows). (**C**,**C’**) Leaf stage spermatids, in which tail MTs compacted (arrows) and the perinuclear MTs cluster at one side of the nucleus (arrowheads). Scale bars: (**A**–**C**), 10 μm; (**B’**,**C’**), 5 μm.

**Figure 3 cells-09-02684-f003:**
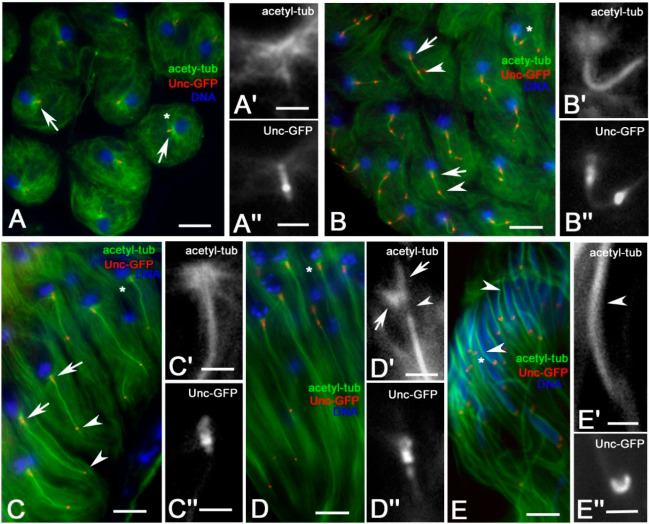
Distribution of acetylated MTs during spermatid elongation from dissected testes. (**A**) Young round spermatids in which centrioles and cilium-like regions are evident (arrows): detail of the small aster of MTs (**A’**) in correspondence of the centriole (**A″**). (**B**,**C**) Elongating spermatids in which the Unc-GFP staining recognized the basal body (arrows) and a subdistal dot (arrowheads) corresponding to the ring centriole: (**B’**,**C’**) are details of the proximal region of the axoneme and of the small aster of MTs close to the nucleus, (**B″**,**C″**) are details of the Unc-GFP localization to the basal body. (**D**) MTs and Unc-GFP labelling in more elongated spermatids: the small aster is no longer visible, but the MTs surround the basal region of the nucleus (arrows). Note the gap between perinuclear and axoneme MTs (arrowhead, **D’**), where the basal body is localized (**D″**). Most of the perinuclear MTs run at one side of the canoe stage nuclei (arrowheads, **E**,**E’**), whereas the Unc-GFP transforms in a compact cluster **(E″**). High magnification images to right details are from spermatids marked by asterisks. Scale bars: (**A**–**D**), 10 μm; (**A’**–**D’**,**A″**–**D″**), 1 μm.

**Figure 4 cells-09-02684-f004:**
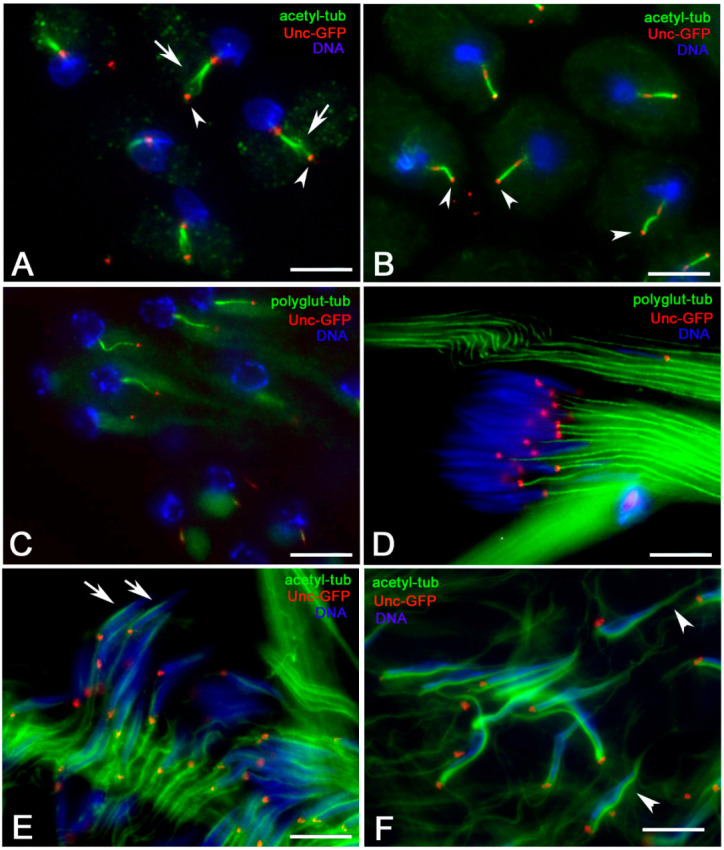
MT stability in elongating spermatids from testis preparations. (**A**) Nocodazole treatment: note the frayed axonemes (arrows) in correspondence of the ring centriole (arrowheads). (**B**) Cold exposure: note the continuity of the axoneme in correspondence of the ring centrioles (arrowheads). Polyglutamylated MTs in young (**C**) and leaf stage (**D**) spermatids: only the axonemal MTs are recognized by the antibody. Perinuclear MTs in control (arrows, (**E**)) and latrunculin-treated (arrowheads, (**F**)) canoe stage nuclei. Scale bars: 10 μm.

**Figure 5 cells-09-02684-f005:**
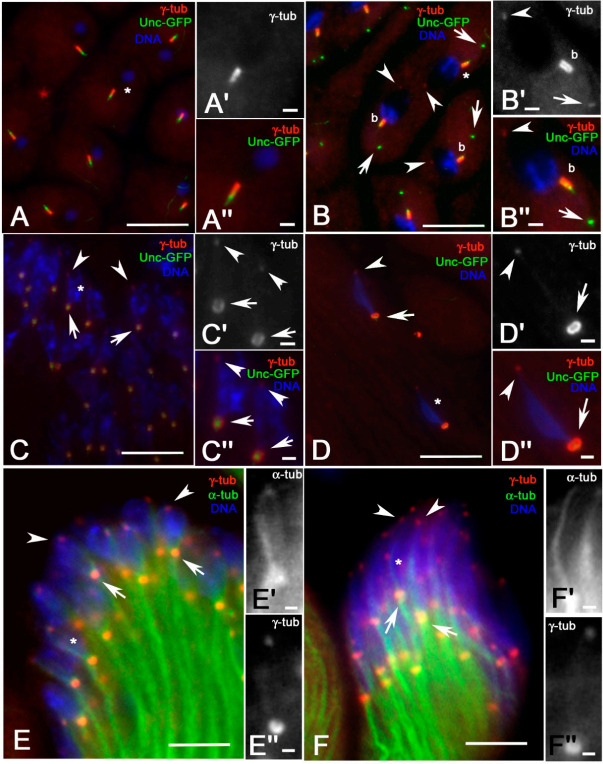
γ-tubulin localization during early spermiogenesis of pupal testes. (**A**–**A″**) Onion stage spermatids: γ-tubulin colocalizes with the basal body. (**B**–**B″**) Young elongating spermatids: γ-tubulin is localized in a small spot (arrowhead) on the apical side of the nucleus, on the basal body (b) and on the ring centriole (arrows). Spermatids at the leaf (**C**–**C″**) and canoe (**D**–**D″**) stages: γ-tubulin forms a circular cluster around the basal body (arrows) and a small spot in the apical region of the nucleus (arrowheads). Counterstain with α-tubulin of leaf (**E**–**E″**) and canoe (**F**–**F″**) stage spermatids shows that γ-tubulin is localized at the base (arrows) and distal ends (arrowheads) of the perinuclear bundles. High magnification images in the right insets are from spermatids marked by asterisks. Scale bars: (**A**–**F**), 7 μm; (**A’**–**F’**,**A″**–**F″**), 0.5 μm.

**Figure 6 cells-09-02684-f006:**
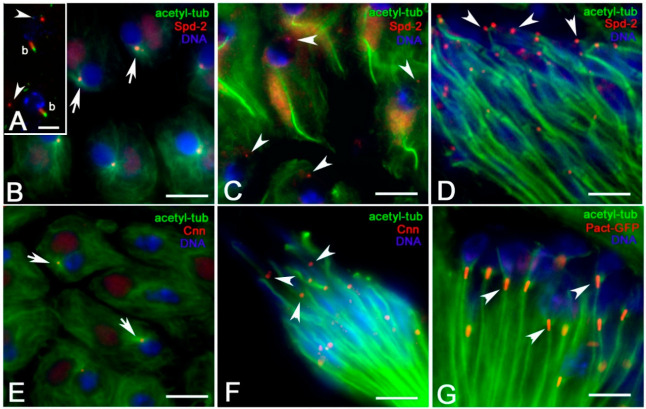
(**A**) Nocodazole-treated *Drosophila* spermatids: γ-tubulin is found as a small spot at one side of the nucleus (arrowheads) and around the basal body (b). Localization of centrosomal proteins involved in γ-tubulin recruitment during early spermiogenesis. Spd-2 is localized to the basal bodies in onion stage spermatids (arrows, (**B**)) and to the apical regions of young elongating (arrowheads, (**C**)) and leaf stage (arrowheads, (**D**)) spermatids. Cnn localizes to the basal bodies in onion stage spermatids (arrows, (**E**)) and centriole adjuncts in leaf stage spermatids (arrowheads, (**F**)). Pact-GFP colocalizes with the centrioles (arrowheads, (**G**). Scale bars: (**A**), 5 μm; (**B**–**G**), 10 μm.

**Figure 7 cells-09-02684-f007:**
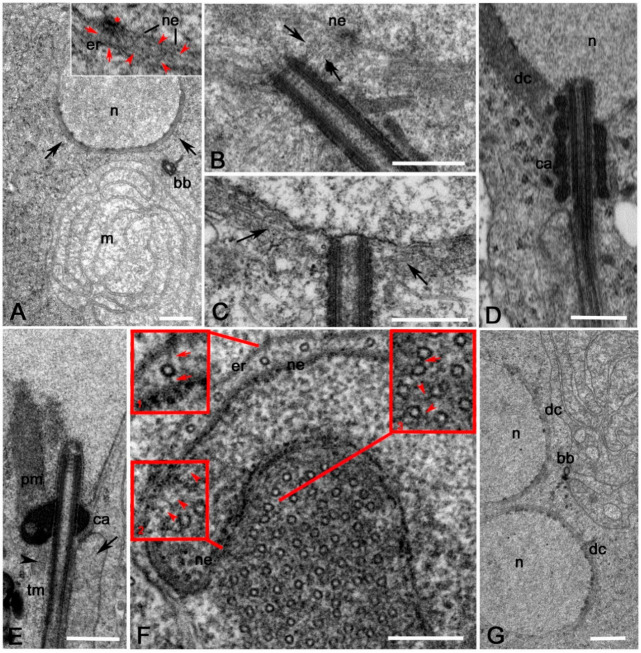
Ultrastructure of early spermiogenesis from dissected pupal testes. (**A**) Onion stage spermatid: m, mitochondria; bb, basal body; n, nucleus; arrows, dense complex. The inset shows detail of the dense complex in which thin filaments (6–7 nm thick) are present running parallel (red arrowheads) to the nuclear envelope (ne) and microfilaments (red arrows) that link nuclear pores (asterisk) to the endoplasmic reticulum (er) that surrounds the basal region of the nucleus. (**B**) Young onion stage spermatid showing that some MTs (arrows) emerging from the basal region of a disoriented basal body contact the nuclear envelope (ne). (**C**) When the basal body docks to the nuclear envelope, most of the MTs emerging from its basal region (arrows) run parallel to the nucleus. (**D**) Detail of a leaf stage spermatid: the pericentriolar material increased to form the centriole adjunct (ca) and the dense complex (dc) moved on the lateral side of the nucleus (*n*). (**E**) Detail of an early canoe stage spermatid in which the basal body is partially enclosed within a nuclear pocket. The centriole adjunct (ca) compacts in a dense cluster that is the anchoring site for the major (arrow) and minor (arrowhead) mitochondrial derivatives, while perinuclear (pm) and tail (tm) MTs originate in opposite direction from the centriole adjunct. (**F**) Cross-section of a canoe stage nucleus (*n*): isolated MTs run along the convex side of the nucleus and some of them show links (red arrows, Inset 1) between the nuclear envelope (ne) and the endoplasmic reticulum (er). Thin filaments (red arrowheads, Inset 2) are visible on the concave side of the nucleus close to the nuclear envelope: the concave side of the nucleus is filled by longitudinal MTs that are often linked together by lateral bridges (red arrow, Inset 3) or show bridges that end within the material (red arrowheads, Inset 3) that form the dense complex. (**G**) *Cog7* mutant spermatid at the onion stage showing two nuclei (*n*) and one basal body (bb): both the nuclei display distinct dense complexes (dc). Scale bars: (**A**–**C, G**), 1 μm; (**D, E**), 0.5 μm; (**F**), 0.1 μm.

**Figure 8 cells-09-02684-f008:**
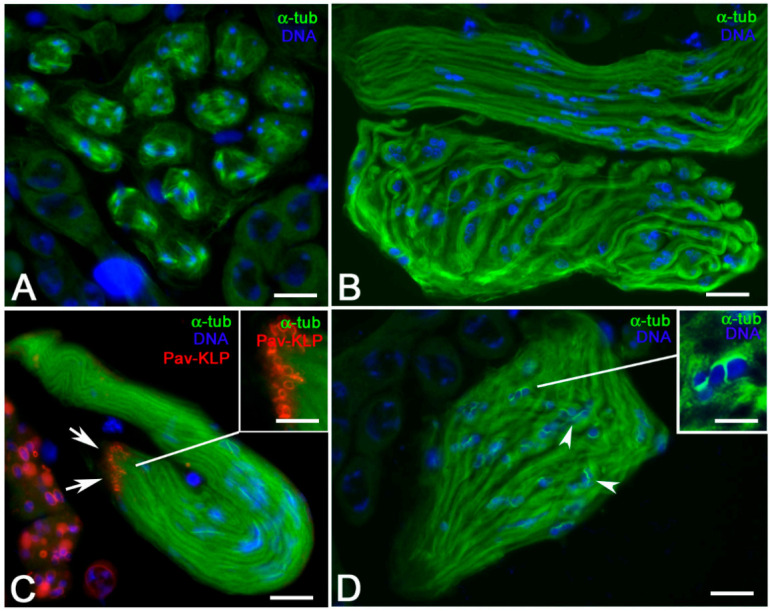
Spermatogenesis and spermiogenesis in *Sas-4* mutant testis. (**A**) Abnormal meiosis with the formation of multiple spindles in primary spermatocytes. (**B**) Despite the absence of basal bodies, the cytoplasm of the elongating spermatids is filled with many MTs. (**C**) Nuclei are scattered within the spermatid cytoplasm, but the ring canals, stained with an antibody against Pavarotti (Pav-KLP), are normally clustered at the end of the spermatid tails (arrows and inset). (**D**) MT bundles are often observed close to the nuclei (arrowheads and inset). Scale bars: (**A**–**D**) 10 μm; insets, 3 μm.

**Figure 9 cells-09-02684-f009:**
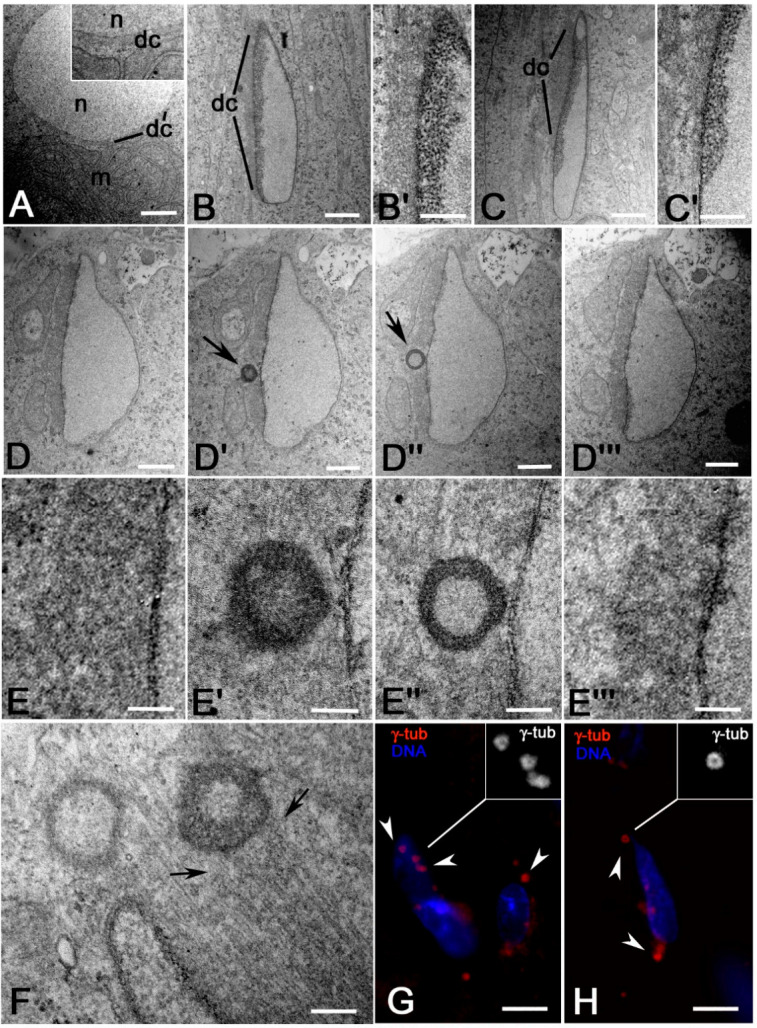
Dense complexes and MTs in *Sas-4* mutant spermatids. (**A**) Onion stage spermatids: note the dense complex (dc) that surrounds the basal region of the nucleus (*n*) in front of the mitochondria (m) (inset). (**B**,**C**) Leaf stage nuclei and respective details (**B’**,**C’**): the dense complex (dc) has variable extension in correspondence of the condensed chromatin. (**D**–**D″’**) Serial longitudinal sections of a leaf stage nucleus and corresponding details (**E**–**E″’**): note the hollow spherical structures (arrows) inside the dense complex. (**F**) Section of the apical region of a leaf stage nucleus showing two hollow spherical structures from which nucleate several parallel MTs (arrows). (**G**,**H**) γ-tubulin localizes around the nuclei as small spots (arrowheads) that appear ring-shaped at higher magnification (insets). Scale bars: (**A**–**C**,**D**–**D″’**), 1 μm; (**B’**,**C’**,**E**–**E″’**,**F**), 0.2 μm; (**G**,**H**) 10 μm.

**Figure 10 cells-09-02684-f010:**
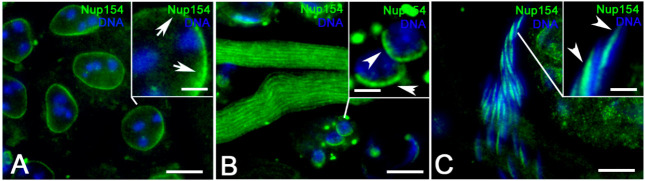
Localization of Nup154 on primary spermatocytes (arrows, (**A**)), in onion stage spermatids (arrowheads, (**B**)) and leaf stage spermatids (arrowheads, (**C**)). Scale bars: (**A**–**C**), 20 μm; (insets) 10 μm.

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
