# Peer review of "The Microtubule Cytoskeleton during the Early Drosophila Spermiogenesis"

_cells, 2020, doi:10.3390/cells9122684_

Round 1

Reviewer 1 Report

The manuscript by Riparbelli and colleagues investigates the organization of the microtubule cytoskeleton during early spermiogenesis in Drosophila. They provide careful characterization, including at the ultrastructural level, of the microtubule cytoskeleton and gamma-tubulin. They report a novel gamma-tubulin focus in the anterior side of the nucleus. They further show that the basal body or centriole adjunct appear to be dispensable for the organization of perinuclear microtubules and propose a role for this new gamma-tubulin center in such process. Although this conclusion in rather indirect, it is a plausible explanation and thus important to be published. I have only few minor comments that require clarification.

1- How do the authors explain the difference in axoneme microtubule morphology (frayed ends) upon nocodazole vs cold treatment?

2- Does the accumulation of gamma-tubulin in the new focus (supposedly acentriolar) uncovered in this paper depend on the presence of microtubules?

3- In the latrunculin-treatment experiments, the authors should measure the length of microtubules associated with the dense body, rather than just providing a qualitative appreciation.

4- The authors refer to gamma-tubulin dynamics but only do experiments in fixed material. It is difficult to infer dynamics in such conditions. Maybe other terminology should be used, such as localization of gamma tubulin at different stages of spermiogenesis?

5- There are few spelling mistakes (e.g. depends by vs. depends on; axoneme microtubule grow vs growth) throughout the manuscript that deserve attention.

Author Response

We thank the referee for his/her comments which allowed us to increase the focus of our manuscript

1- How do the authors explain the difference in axoneme microtubule morphology (frayed ends) upon nocodazole vs cold treatment?

We report this sentence to explain the different microtubule morphology upon nocodazole vs cold treatment: “To assess MT stability during early spermiogenesis, we treated Drosophila testes with the tubulin-sequestering drug nocodazole and with cold temperature. Nocodazole has the advantage to being more specific for MTs, whereas cold temperature is likely to affect other cellular processes [28]. Both treatments produced the loss of cytoplasmic and ciliary cap MTs suggesting that the majority of the cytoplasmic MTs recognized by the antibody against acetylated tubulin were less stable. Remarkably, the distal end of the axonemal MTs detached from the ring centrioles, and frayed after nocodazole incubation (Figure 4A), whereas cold temperature did not affect the axoneme integrity (Figure 4B). The different results could be explained by factors that stabilize MTs against cold-induced depolymerization at the ring centrioles, but more sensitivity to drug treatment thus resulting in frayed distal ends of the axonemes.”

2- Does the accumulation of gamma-tubulin in the new focus (supposedly acentriolar) uncovered in this paper depend on the presence of microtubules?

In agreement with the comment of the Referee we look at the localization of gamma-tubulin after microtubule depolymerization. We report our observations with this sentence: “We then wondered whether the gamma-tubulin accumulation persisted in young spermatids after depolymerization of the cytoplasmic MTs. Thus, we counterstained nocodazole treated spermatids expressing Unc-GFP with an antibody against gamma-tubulin. We find that gamma-tubulin accumulation persisted at the basal bodies and as small dots at the apical side of the spermatid nuclei (Figure 6A). Therefore, the localization of gamma-tubulin during early spermiogenesis did not strictly depend by MT integrity.”

3- In the latrunculin-treatment experiments, the authors should measure the length of microtubules associated with the dense body, rather than just providing a qualitative appreciation.

Accordingly, we measured the length of the perinuclear microtubules in control and latrunculin treated leaf stage spermatids. We added this paragraph: “We find that the perinuclear MT bundles measured 12.97±01.4 mm in control canoe stage spermatids (Figure 4E, n=47) and reached 23.61±0.21 mm in drug treated canoe stage spermatids (Figure 4F, n=38).”

4- The authors refer to gamma-tubulin dynamics but only do experiments in fixed material. It is difficult to infer dynamics in such conditions. Maybe other terminology should be used, such as localization of gamma tubulin at different stages of spermiogenesis?

The referee's opinion is correct: we cannot use dynamics for fixed material. Accordingly we changed gamma-tubulin dynamics with localization.

5- There are few spelling mistakes (e.g. depends by vs. depends on; axoneme microtubule grow vs growth) throughout the manuscript that deserve attention.

We corrected the spelling errors throughout the manuscript. We apologize for these mistakes

Reviewer 2 Report

In the manuscript, ‘The microtubule cytoskeleton during the early Drosophila spermiogenesis’ by Riparbelli et al, submitted to the journal ‘Cells’, the authors newly described the localization of γ-tubulin at the anterior pole of the nucleus early spermatids. This itself is a very interesting finding. Though the manuscript is largely descriptive, this area of research could benefit from this data.

Here are my concerns which needs to be addressed before I can recommend this manuscript for publication in the journal ‘Cells’.

>> Major concern

My most important concern:  The authors show that the perinuclear microtubules and the dense complex are present in spermatids lacking centrioles (in Sas4 spermatids; Fig 7 and Fig 8). Therefore, they claim that the basal body or the centriole adjunct seem to be dispensable for the organization and assembly of these structures. To directly address the role of the newly identified apical γ-tubulin (and Spd-2) dependent organization of perinuclear microtubules and the dense complex. I think it makes a lot of sense to check for γ-tubulin (and Spd-2) localization in early Sas4 spermatids.

>> Minor concerns

Fig.4: Scale bar description for E-E’’ and F-F’’ are missing

>Line numbers and corrections suggested….

42: ‘growths’ to ‘grows’

163: ‘week’ to ‘weak’

171: ‘ster’ to ‘aster’

184: ‘exposition’ to ‘exposure’

260: ‘are likely involved in assist basal body’ to ‘likely assist basal body’

325: The full stop that appeared in the middle of the sentence after ’Pavarotti (Pav-KLP).’ has to be removed

393-394: ‘However, the CnnT isoform has not been detected within the centriole adjunct and at the apical end of the nuclei.’ --- Citation is missing for this statement

405-406: ‘depend by’ the to ‘depend on the’

432-433: ‘However, the dense complex of Sas-4 elongating spermatids does not support full nuclear shaping and elongated nuclei were rarely found.’ --- This sentence is not clear

Additionally, I recommend a thorough review of the manuscript text to improve your writing.

Author Response

We thank the Reviewer for his/her positive evaluation of our manuscript. Here is the point-by-point answer to the Reviewer’s comments:

My most important concern: The authors show that the perinuclear microtubules and the dense complex are present in spermatids lacking centrioles (in Sas4 spermatids; Fig 7 and Fig 8). Therefore, they claim that the basal body or the centriole adjunct seem to be dispensable for the organization and assembly of these structures. To directly address the role of the newly identified apical γ-tubulin (and Spd-2) dependent organization of perinuclear microtubules and the dense complex. I think it makes a lot of sense to check for γ-tubulin (and Spd-2) localization in early Sas4 spermatids.

As required, we made the localization of γ-tubulin on early Sas-4 spermatids (new Fig 9G,H). We confirmed here that γ-tubulin is present in the absence of basal bodies. This was a very useful suggestions: surprisingly, the γ-tubulin signal appears as small rings that resemble the dense bodies found by electron microscopy.

Fig.4: Scale bar description for E-E’’ and F-F’’ are missing

We add the scale bars.

Line numbers and corrections suggested….

We made all the corrections as required

393-394: ‘However, the CnnT isoform has not been detected within the centriole adjunct and at the apical end of the nuclei.’ --- Citation is missing for this statement

We apologize for these mistakes. We added the reference here.

432-433: ‘However, the dense complex of Sas-4 elongating spermatids does not support full nuclear shaping and elongated nuclei were rarely found.’ --- This sentence is not clear

We removed this sentence

Reviewer 3 Report

The major issues with the paper are accessibility to a non-specialist audience, the lack of clear hypotheses and the relevance to a non-specialist audience.

There may be interesting and novel findings in the submitted manuscript that are of broad interest, but the authors fail to make a compelling case. There is some beautiful immunofluorescence images but more needs to done for their significance to be apparent. The results are largely descriptive, which is consistent with a title that does not make any claim.

Some examples of why a non-specialist found this paper difficult to comprehend:

  • The introduction and results seem to assume a great deal of background knowledge on the model system. A schematic diagram at the start explaining the key stages and localization of key factors would be very useful.
  • Terms such as onion, leaf and canoe stage need to be introduced earlier than in Figure legends.
  • The reader should not have to go to the methods to understand that the micrographs presented are dissected testes (rather than explants grown in vitro, or adult sections, for example).
  • What is unc-GFP? I couldn't find any explanation as to what unc is, or what structure it labels. It is therefore impossible to appreciate why it is remarkable that the protein is localized to "three distinctive regions".
  • There are substantial language and phrasing issues that are a barrier for the audience: "The dense complex appears at one side of the young spermatid nuclei in correspondence of the basal body as a cluster of dense material enclosed between the nuclear envelope and an elongated cisterna of endoplasmic reticulum"; "Although, the actin cytoskeleton has been implicated in nuclear dynamics, the treatment with latrunculin that affects microfilament dynamics does not give appreciable effects on the nuclear shaping but impacts on the microtubules of the dense complex that significantly elongate"; "This is also supported by the observation that a dense hemispherical nuclear cup is present at the onion stage in Sas-4 mutant spermatids and in Cog7 meiotic mutant spermatids containing multiple nuclei not accompanied by as much basal bodies".

Minor issues

32-32 – This assertion requires a reference and/or rephrasing to indicate that this is a generally accepted hypothesis in the literature, but observation of ribosomes within the apical region of the ciliary cap suggests protein synthesis in this region could assist in sustaining microtubule elongation, as opposed to elongation being sustained purely with tubulin translocated by diffusion.

36-37 – The paragraph change here needs a linking sentence of some description. It’s not clear how the information in each paragraph connects.

53 – Reference needed. This sentence seems to be intended to establish contrast between spermatid elongation and shaping of the nucleus, but is not structured correctly to achieve that. E.g. Axonemes and centriole not required for elongation, in contrast to nuclear shaping.

48 – 52 – This paragraph needs further explanation. Currently it’s not sufficiently clear how the ideas in these two sentences are linked.

68 – Which aspects? This sentence would benefit from appearing earlier in the introduction, and from further elaboration.

Methods – catalogue numbers or other specific product IDs should be provided, particularly for all antibodies used. It is not possible to reproduce the experiments based on the information provided.

111 – 114 - Microscope acquisition software name and version number should be provided here.

Figures – 7/9 Show higher magnification cutouts of regions of interest superimposed over their source image, while 1/2/4 appear to show similar cutouts presented alongside their source images. All figures should use the same style, and cutouts should show the region on the source image that they are drawn from. Some figures also use a single scale bar to indicate scale for all panels, while others provide a scale bar for each individual image.

125–141 – This purpose of this section appears to be to characterize the localization and structure of microtubules in spermatids at a number of developmental stages. This needs to be clearly stated, and the conclusions drawn here framed in that context.

140 – This sentence appears to be referencing Figure 2, but does not cite it.

182 – Why were these two methods chosen?

143-147 – Features indicated by arrows should either be explained inside brackets, or outside brackets. Not both.

144 – 147 – In reference to C’, B’: Arrows/arrowheads are used on a single channel image to draw attention to the spatial relationship between nuclei and microtubule clusters. Excluding the channel showing DNA makes it more complicated for a reader to understand the relative positions of these features.

151 – Unc protein, and its function and relevance in this context, should be explained here. 

265 – Not sufficiently clear that the authors are talking about one structure (pericentriolar material) that has now transitioned into another (centriole adjunct). This sentence reads as though the term for the structure is being redefined, rather than the structure itself undergoing a morphological transition.

201 / 301 – Results sub-headings should assert the main conclusion of the data in each section. Some sub-headings are expressed this way (301), while others are not (201).

479 – A concluding statement that summarises the most important findings in relation to the poorly understood aspects of microtubule dynamics identified in the introduction would help to communicate the impact of the paper.

Author Response

We thank the referee for his/her comments which allowed us to make the manuscript more attractive to read. Here is the point-by-point answer to the Reviewer’s comments

Some examples of why a non-specialist found this paper difficult to comprehend:

  • The introduction and results seem to assume a great deal of background knowledge on the model system. A schematic diagram at the start explaining the key stages and localization of key factors would be very useful.

We added a schematic representation of the first phases of Drosophila spermiogenesis (Figure 1) showing the different stages of spermatid elongation and the main structural components of the cells.

  • Terms such as onion, leaf and canoe stage need to be introduced earlier than in Figure legends.

We introduced the stages of spermiogenesis in the first section of the Results

  • The reader should not have to go to the methods to understand that the micrographs presented are dissected testes (rather than explants grown in vitro, or adult sections, for example).

We added in some figure legends that the pictures were from dissected testes.

  • What is unc-GFP? I couldn't find any explanation as to what unc is, or what structure it labels. It is therefore impossible to appreciate why it is remarkable that the protein is localized to "three distinctive regions".

As required, we add this sentence at the beginning of result section “Therefore, we stained Drosophila testes expressing Uncoordinated (Unc)-GFP, a potential ortholog of the orofaciodigital syndrome 1 (Ofd1) that is required for ciliogenesis and mechanosensation in Drosophila and localizes to the basal body [19] with an antibody against acetylated tubulin to recognize both centrioles and stable MTs associated with the axonemes”

  • There are substantial language and phrasing issues that are a barrier for the audience: "The dense complex appears at one side of the young spermatid nuclei in correspondence of the basal body as a cluster of dense material enclosed between the nuclear envelope and an elongated cisterna of endoplasmic reticulum";

We rephrased this sentence: “The dense complex is firstly found in young spermatids as a hemispherical thin cup of electrondense material on the side of the nucleus opposite to the nebenkern.”

  • "Although, the actin cytoskeleton has been implicated in nuclear dynamics, the treatment with latrunculin that affects microfilament dynamics does not give appreciable effects on the nuclear shaping but impacts on the microtubules of the dense complex that significantly elongate";

The sentence has been modified: “Microfilaments have been implicated in nuclear shaping during Drosophila spermiogenesis [13]. Therefore, treatment with the actin inhibitor latrunculin would affect the nuclear shape of young spermatids. However, we find that this treatment does not give appreciable effects on the nuclear shaping but impacts on the MTs of the dense complex that significantly elongate.”

  • "This is also supported by the observation that a dense hemispherical nuclear cup is present at the onion stage in Sas-4 mutant spermatids and in Cog7 meiotic mutant spermatids containing multiple nuclei not accompanied by as much basal bodies".

The paragraph has been modified “This is also supported by the finding that the dense complex assembles during normal spermiogenesis before the basal body contacted the nuclear envelope. Moreover, several Drosophila meiotic mutants display spermatids that contain two or more nuclei with dense complexes but only one basal body”.

32-32 – This assertion requires a reference and/or rephrasing to indicate that this is a generally accepted hypothesis in the literature, but observation of ribosomes within the apical region of the ciliary cap suggests protein synthesis in this region could assist in sustaining microtubule elongation, as opposed to elongation being sustained purely with tubulin translocated by diffusion.

The paragraph has been modified as suggested by the Reviewer’s comment: “The assembly and further elongation of the axonemal MTs would require tubulin translocation by diffusion through the ciliary cap. However, the observation of ribosomes within the apical region of the ciliary cap [6] suggests that protein synthesis in this region could assist in sustaining MT elongation.”

36-37 – The paragraph change here needs a linking sentence of some description. It’s not clear how the information in each paragraph connects.

To link the paragraphs, we add this sentence: “The centriole that organize the sperm axoneme is associated in young spermatids with a low amount of pericentriolar material from which a small astral array of MT nucleates.”

53 – Reference needed. This sentence seems to be intended to establish contrast between spermatid elongation and shaping of the nucleus, but is not structured correctly to achieve that. E.g. Axonemes and centriole not required for elongation, in contrast to nuclear shaping.

We modified the sentence and added two references: “The MTs that nucleate from the centriole adjunct have been involved in the shaping of the spermatid nucleus [6,13]”.

48 – 52 – This paragraph needs further explanation. Currently it’s not sufficiently clear how the ideas in these two sentences are linked.

We modified the sentence to link the two paragraphs: It has been suggested that spermatid elongation in Drosophila requires the cooperation between cytoplasmic non-axonemal MTs and the mitochondrial derivatives that might provide a structural scaffold for MT reorganization as inferred by the finding of g-tubulin spots on the mitochondrial membrane [11].

68 – Which aspects? This sentence would benefit from appearing earlier in the introduction, and from further elaboration.

We think better to remove this sentence

Methods – catalogue numbers or other specific product IDs should be provided, particularly for all antibodies used. It is not possible to reproduce the experiments based on the information provided.

We agree with the referee’s comment and we added the catalogue numbers as required.

111 – 114 - Microscope acquisition software name and version number should be provided here.

We added the name of the software.

Figures – 7/9 Show higher magnification cutouts of regions of interest superimposed over their source image, while 1/2/4 appear to show similar cutouts presented alongside their source images. All figures should use the same style, and cutouts should show the region on the source image that they are drawn from. Some figures also use a single scale bar to indicate scale for all panels, while others provide a scale bar for each individual image.

We agree with the referee’s comment. Figures should use the same style, but we are forced to represent details in different format. For example, in Figures 2 and 4 we organized details on the side of the main panels, since we were unable to put these large insets within the source images. Here, we mark with asterisks the spermatids that have been magnified. By contrast, in Figures 7 and 9 we put insets within the main figures since we didn't need to add more channels. In this case we indicated the details that have been magnified.

As suggested, we added the missing scale bars. We apologize for this mistake.

125–141 – This purpose of this section appears to be to characterize the localization and structure of microtubules in spermatids at a number of developmental stages. This needs to be clearly stated, and the conclusions drawn here framed in that context.

We added this sentence at the beginning of the section: “Since MTs represent an important actor during early morphogenesis of the Drosophila spermatids, we examined the distribution of these cytoskeletal organelles at different stages of early spermiogenesis (Figure 1) to better understand their role in nuclear shaping and tail elongation.”

140 – This sentence appears to be referencing Figure 2, but does not cite it.

Thanks, we added the Figure citation.

182 – Why were these two methods chosen?

We added this paragraph “To assess MT stability during early spermiogenesis, we treated Drosophila testes with the tubulin-sequestering drug nocodazole and with cold temperature. Nocodazole has the advantage to being more specific for MTs, whereas cold temperature is likely to affect other cellular processes [28].”

143-147 – Features indicated by arrows should either be explained inside brackets, or outside brackets. Not both.

The legends have been corrected accordingly.

144 – 147 – In reference to C’, B’: Arrows/arrowheads are used on a single channel image to draw attention to the spatial relationship between nuclei and microtubule clusters. Excluding the channel showing DNA makes it more complicated for a reader to understand the relative positions of these features.

We agree with the Reviewer about the importance of a picture in which both DNA and tubulin would be useful to understand the reciprocal localization. As suggested, we add the DNA channel to tubulin, but as you can see from the picture I enclosed, the details of the perinuclear MTs are more difficult to see. I agree with the Referee’s opinion that also the DNA channel could be useful, but at this moment it is difficult for me to make a final decision. Moreover, some nuclei are at different focal planes than MTs, since I focused on MTs. I would be very grateful for any opinion and suggestion in this regard.

151 – Unc protein, and its function and relevance in this context, should be explained here. 

We added this sentence “The Unc protein is involved in centriole-to-basal body conversion and is specifically expressed by spermatocyte and spermatid centrioles that nucleate a ciliary axoneme [19]. We find that this protein was localized in three distinct regions of the basal bodies….”

265 – Not sufficiently clear that the authors are talking about one structure (pericentriolar material) that has now transitioned into another (centriole adjunct). This sentence reads as though the term for the structure is being redefined, rather than the structure itself undergoing a morphological transition.

We changed the sentence: “The dense material that surrounded the basal body increased in elongating spermatids at the leaf stage and formed a compact cluster, called the centriole adjunct (Figure 7D).”

201 / 301 – Results sub-headings should assert the main conclusion of the data in each section. Some sub-headings are expressed this way (301), while others are not (201).

We agree with the Referee’s comments and changed the first sub-heading of Result section: “MT distribution significantly changes during early spermiogenesis”

479 – A concluding statement that summarises the most important findings in relation to the poorly understood aspects of microtubule dynamics identified in the introduction would help to communicate the impact of the paper.

We added this paragraph as concluding statement: We have sought to better characterize the MT cytoskeleton during different stages of the early Drosophila spermiogenesis. Our observations uncover a new g-tubulin localization at the apical end of the nucleus. This g-tubulin focus is likely involved in the nucleation of a population of perinuclear MTs that fill the dense complex. Such g-tubulin foci are also found in Sas-4 mutant spermatids where they nucleate distinct bundles of perinuclear MTs. Remarkably, the Sas-4 mutant spermatids lack basal bodies, but have distinct dense complexes suggesting that centrioles or/and centriole adjuncts are dispensable for the assembly of this structure. Future studies aimed at addressing the regulation of MT nucleation and dynamics during early Drosophila spermiogenesis are required to understand whether MTs specifically contribute to nuclear shaping and sperm tail elongation.

Reviewer 4 Report

The paper of Riparbelli et al., contains new and important data on the role of microtubules in Drosophila spermiogenesis. At the same time, there are some shortcomings and typing errors in the text of the paper, which must be corrected so that the article can be published in a special issue "Centrosome" of the Cells journal.

Line 24 and further throughout the text

It may be better to use the abbreviations MT and MTs instead of microtubule and microtubules.

Line 35

In the text:

“This finding suggests that some protein synthesis cold occur in this region to sustain microtubule elongation”.

Correction:

“This finding suggests that some protein synthesis could occur in this region to sustain microtubule elongation”.

Line 40

I propose to add this sentence, because for scientists who work with mammalian, centriolar adjunct is other type of structure.  

“It must be noted that the term centriolar adjunct in insect spermatids denotes a completely different structure than in mammalian spermatids, where this term denotes a structure growing from the proximal centriole and similar in morphology to the primary cilium (Alieva et al., 2018; Garanina et al., 2019).”

Alieva, I.B.; Staub, C.; Uzbekova, S.; Uzbekov, R.E. A question of flagella origin for spermatids – mother or daughter centriole? In Flagella and cilia. Types structure and functions, Uzbekov, R.E., Ed. Nova Science Publishers, Inc.: New York, 2018; pp 109–126.

Garanina, A.S.; Alieva, I.B.; Bragina, E.E.; Blanchard, E.; Arbeille, B.; Guerif, F.; Uzbekova, S.; Uzbekov, R.E. The centriolar adjunct -appearance and disassembly in spermiogenesis and the potential impact on fertility. Cells 2019, 8.

Line 116

In the text:

at 4 C  

Correction:

at 4°C

Line 149

I think you can add here reference to more early publication too:

“Meiosis I centrosomes and mitotic centrosomes both have two centrioles. One feature that is unique to meiosis II in Drosophila spermatogenesis is that a single centriole exists in each centrosome. “

Li, K.; Xu, E. Y.; Cecil, J. K.; Turner, F. R.; Megraw, T. L.; Kaufman, T. C. Drosophila centrosomin protein is required for male meiosis and assembly of the flagellar axoneme. 1998. J. Cell biology, 141(2), 455–467. https://doi.org/10.1083/jcb.141.2.455. 

Line 162

In the text:

Specifically, we observed a proximal Unc-GFP signal that was week at the beginning of 163 axoneme  elongation  (Figure  2B’’)

Correction:

Specifically, we observed a proximal Unc-GFP signal that was weak at the beginning of 163 axoneme  elongation  (Figure  2B’’)

Line 171

In the text:

“detail of the small ster of”

Correction:

“detail of the small aster of”

Line 196

Recent studies have shown that the nucleation of microtubules and actin filaments is closely related and the centrosome can also be the nucleation center of actin filaments.

You can add references here:

Inoue, D.; Obino, D.; Pineau, J.; Farina, F.; Gaillard, J.; Guerin, C.; Blanchoin, L.; Lennon-Dumenil, A.M.; Thery, M. Actin filaments regulate microtubule growth at the centrosome. EMBO J. 2019, 38, e99630.

Uzbekov, R.E.; Avidor-Reiss, T. Principal Postulates of Centrosomal Biology. Version 2020. Cells 2020, 9, 2156.

Line 202

You can add references here:

“Due to the main role of γ-tubulin in microtubule nucleation (Oakley and Oakley, 1989; Joshi,et al., 1992)”.  

Oakley, C.E.; Oakley, B.R. Identification of gamma-tubulin, a new member of the tubulin superfamily encoded by mipa gene of aspergillus nidulans. Nature 1989, 338, 662–664.

Joshi, H.C.; Palacios, M.J.; McNamara, L.; Cleveland, D.W. Gamma-tubulin is a centrosomal protein required for cell cycle-dependent microtubule nucleation. Nature 1992, 356, 80–83.

Line 281 and line 293

«are present microfilaments running parallel (red arrowheads) to the nuclear envelope (ne) and microfilaments»

The term “microfilaments” is usually used for actin filaments. Are these filaments actin filaments?

Line 305

I did not find (Figure  6H) in the paper. 

Line 312

Sas4  mutant testis ?

Lines 318, 333, 334, 335, 343

Sometimes you used the term “Sas-4 mutant spermatids”, sometimes “Sas-4 spermatids” and “Sas-4 mutant".

 It is not clear. Please use identical definition for mutant in all cases.   

Line 330

“Such microtubules were short and stop just beyond the nuclei”

May be better:

“These microtubules were short and ended just beyond the nuclei.”

In the Discussion, it would be interesting to read more about the comparison of insect and mammalian spermiogenesis. In particular, why is there only one centriole in insect spermatids, and two in mammalian spermatids? What is the analogue of the centriolar adjunct in mammals? (segmented columns?) Why in insects the centriole first contacts the nucleus, and then the axoneme grows to the cell membrane, while in vertebrates, on the contrary, first the basal body (distal centriole) contacts the cell membrane and forms a flagellum and only then binds to the nucleus?

But these are just my optional wishes.

Author Response

We thank the Reviewer for his/her positive evaluation of our manuscript. Here is the point-by-point answer to the Reviewer’s comments:

Line 24 and further throughout the text - It may be better to use the abbreviations MT and MTs instead of microtubule and microtubules.

As suggested, we used the abbreviations (MTs)

Line 35 -In the text: “This finding suggests that some protein synthesis cold occur in this region to sustain microtubule elongation”. Correction: “This finding suggests that some protein synthesis could occur in this region to sustain microtubule elongation”.

We made the correction

Line 40 I propose to add this sentence, because for scientists who work with mammalian, centriolar adjunct is other type of structure.  

“It must be noted that the term centriolar adjunct in insect spermatids denotes a completely different structure than in mammalian spermatids, where this term denotes a structure growing from the proximal centriole and similar in morphology to the primary cilium (Alieva et al., 2018; Garanina et al., 2019).”

 We added the sentence suggested and the references herein.

I think you can add here reference to more early publication too: “Meiosis I centrosomes and mitotic centrosomes both have two centrioles. One feature that is unique to meiosis II in Drosophila spermatogenesis is that a single centriole exists in each centrosome. “

We added this paragraph and the reference suggested.

Several typos have been corrected through the text as suggested.

Line 196 Recent studies have shown that the nucleation of microtubules and actin filaments is closely related and the centrosome can also be the nucleation center of actin filaments.

We added the sentence and the references suggested.

Line 202 You can add references here: “Due to the main role of γ-tubulin in microtubule nucleation (Oakley and Oakley, 1989; Joshi,et al., 1992)”.  

We added the references suggested

Line 281 and line 293 «are present microfilaments running parallel (red arrowheads) to the nuclear envelope (ne) and microfilaments» The term “microfilaments” is usually used for actin filaments. Are these filaments actin filaments?

In agreement with the referee’s comment we are unable to affirm that these filaments are microfilaments unless a biochemical analysis. Therefore, we removed the term microfilaments and added “…thin filaments (6-7 nm thick)..”.

Line 305 I did not find (Figure 6H) in the paper.

Sorry for the mistake. We corrected the number. 

Line 312 Sas4 mutant testis ? Lines 318, 333, 334, 335, 343 Sometimes you used the term “Sas-4 mutant spermatids”, sometimes “Sas-4 spermatids” and “Sas-4 mutant". It is not clear. Please use identical definition for mutant in all cases.

We made the corrections accordingly

Line 330

“Such microtubules were short and stop just beyond the nuclei” May be better: “These microtubules were short and ended just beyond the nuclei.”

We made the correction.

In the Discussion, it would be interesting to read more about the comparison of insect and mammalian spermiogenesis. In particular, why is there only one centriole in insect spermatids, and two in mammalian spermatids? What is the analogue of the centriolar adjunct in mammals? (segmented columns?) Why in insects the centriole first contacts the nucleus, and then the axoneme grows to the cell membrane, while in vertebrates, on the contrary, first the basal body (distal centriole) contacts the cell membrane and forms a flagellum and only then binds to the nucleus? But these are just my optional wishes.

The discussion topics suggested by the Referee are very interesting. At this moment we are writing a review focused on these aspects.

Round 2

Reviewer 2 Report

All of my previous concerns were addressed.

Best regards

Reviewer 3 Report

The authors have improved the manuscript but the changes do not change my opinion of a lack of a clear hypothesis and the results being largely descriptive. I am not convinced a compelling case has been made for any kind of general interest for this work.